# Chromosome and Genome Divergence between the Cryptic Eurasian Malaria Vector-Species *Anopheles messeae* and *Anopheles daciae*

**DOI:** 10.3390/genes11020165

**Published:** 2020-02-05

**Authors:** Anastasia N. Naumenko, Dmitriy A. Karagodin, Andrey A. Yurchenko, Anton V. Moskaev, Olga I. Martin, Elina M. Baricheva, Igor V. Sharakhov, Mikhail I. Gordeev, Maria V. Sharakhova

**Affiliations:** 1Department of Entomology and the Fralin Life Science Institute, Virginia Polytechnic and State University, 360 West Campus Drive, Blacksburg, VA 24061, USA; naumenko@vt.edu (A.N.N.); andreyurch@gmail.com (A.A.Y.); oshar@vt.edu (O.I.M.); igor@vt.edu (I.V.S.); 2Laboratory of Evolutionary Genomics of Insects, the Federal Research Center Institute of Cytology and Genetics, Siberian Branch of the Russian Academy of Sciences, 10 Prospekt Lavrentyeva, 630090 Novosibirsk, Russia; karagodin@bionet.nsc.ru; 3Department of General Biology and Ecology, Moscow Regional State University, 10a Radio Street, 105005 Moscow, Russia; av.moskaev@mgou.ru (A.V.M.); gordeev_mikhail@mail.ru (M.I.G.); 4Laboratory of Cell Differentiation Mechanisms, the Federal Research Center Institute of Cytology and Genetics, Siberian Branch of the Russian Academy of Sciences, 10 Prospekt Lavrentyeva, 630090 Novosibirsk, Russia; barich@bionet.nsc.ru; 5Laboratory of Ecology, Genetics and Environment Protection, Tomsk State University, 36 Lenina Street, 634041 Tomsk, Russia

**Keywords:** malaria mosquitoes, polymorphic inversions, genome, internal transcribed spacer 2

## Abstract

Chromosomal inversions are important drivers of genome evolution. The Eurasian malaria vector *Anopheles*
*messeae* has five polymorphic inversions. A cryptic species, *An. daciae*, has been discriminated from *An. messeae* based on five fixed nucleotide substitutions in the internal transcribed spacer 2 (ITS2) of ribosomal DNA. However, the inversion polymorphism in *An. daciae* and the genome divergence between these species remain unexplored. In this study, we sequenced the ITS2 region and analyzed the inversion frequencies of 289 *Anopheles* larvae specimens collected from three locations in the Moscow region. Five individual genomes for each of the two species were sequenced. We determined that *An. messeae* and *An. daciae* differ from each other by the frequency of polymorphic inversions. Inversion X1 was fixed in *An. messeae* but polymorphic in *An. daciae* populations. The genome sequence comparison demonstrated genome-wide divergence between the species, especially pronounced on the inversion-rich X chromosome (mean Fst = 0.331). The frequency of polymorphic autosomal inversions was higher in *An. messeae* than in *An. daciae.* We conclude that the X chromosome inversions play an important role in the genomic differentiation between the species. Our study determined that *An. messeae* and *An. daciae* are closely related species with incomplete reproductive isolation.

## 1. Introduction

Chromosomal inversions are essential drivers of genome evolution in diploid organisms [1]. When a chromosomal inversion occurs, a piece of the chromosome flips 180 degrees and produces a reverse order of the genetic material. As a result, this part of the genome becomes resistant to recombination during meiosis. By capturing different combinations of alleles, inversions have effects on ecological, behavioral, and physiological adaptations of the species to the natural environment [2]. The role of the chromosomal inversions in adaptation and evolution of different *Drosophila* species has been studied for several decades [3,4,5]. More recently, genome sequencing and mapping revealed functionally important inversions in the human genome [6,7,8,9,10]. Polymorphic inversions have been shown to be associated with epidemiologically important phenotypes in Afrotropical populations of *Anopheles* [11,12,13,14]. Understanding the population genetics and mechanisms of taxa diversification is extremally important for species that transmit human diseases [15]. Among other malaria mosquitoes, *An. messeae* Falleroni is one of the most geographically widespread [16] and genetically diverse [17] species of malaria mosquitoes in Eurasia. Distribution of the species extends from Ireland in the West to the Amur river region in the East and from Scandinavia and Yakutia in the North to Iran and Northern China in the South [18]. *An. messeae* was the primary malaria vector in Russia [19] and now represents a threat to malaria re-emergence in the Northern territories of Eurasia because of global climate change [20,21]. This species is susceptible to *Plasmodium vivax* but not to the tropical *P. falciparum* malaria parasite [16].

*An. messeae* was originally described as a subspecies of *An. maculipennis* within the *Anopheles maculipennis* complex [22]. According to current systematics, the Maculipennis group of malaria mosquitoes comprises eleven Palearctic species: *An. artemievi*, *An. atroparvus*, *An. beklemishevi*, *An. daciae*, *An. labranchiae*, *An. maculipennis*, *An. martinius*, *An. melanoon*, *An. messeae*, *An. persiensis*, and *An. sacharovi* [23]. Species in this group have traditionally been recognized based on the chorion patterns of their eggs [24,25] and the banding patterns of their polytene chromosomes [17,26,27,28,29]. Three species—*An. artemievi* [30], *An. persiensis* [31], and *An. daciae* [32]—were identified more recently using nucleotide sequence substitutions in the Internal Transcribed Spacer 2 (ITS2) of the ribosomal DNA (rDNA). *An. daciae*, collected near the Danube river in Romania, was discriminated from *An. messeae* based on five fixed nucleotide substitutions in ITS2 and by egg morphology; the eggs of *An. daciae* are generally darker and smaller and have tubercles that are organized in patches of a slightly different shape [32]. Using the molecular approach, *An. daciae* was later discovered in Germany [33,34,35], England and Wales [36], and more recently in Poland [37], the Czech republic, Slovakia [38], and Serbia [39]. In most locations, *An. daciae* was found to be in sympatry with its cryptic species *An. messeae*. However, the detailed analysis of the inversion polymorphism in *An. daciae* and the genetic mechanisms of the genomic diversification of these two species remain largely unexplored.

Five highly polymorphic chromosomal inversions have been described in *An. messeae* species: X1, X2, and X4 on chromosome X; 2R1 on chromosome 2; 3R1 and 3L1 on different arms of chromosome 3 [40]. A latitude cline was described for the 2R1 inversion where the inverted variant was more abundant in northern populations, suggesting that this inversion could be involved in adaptation to cold temperatures and success in overwintering [41]. The frequencies of the X1 and 3R1 inversions displayed a West-East longitude cline with higher frequencies of the inverted variants found in Eastern populations. The X2 inversion is endemic in Western Siberian populations [40]. Comparison of the inversion polymorphisms among populations also suggested that area-specific biological or behavioral adaptations were likely have occurred [42,43]. Inversion frequencies have been shown to be variable in different water reservoirs located next to each other, suggesting involvement of the inversions in local adaptations at the subpopulation level [44]. Inversion frequencies also varied during the summer period with standard variants more abundant in the middle of the summer [45]. Although, inversion frequencies did not significantly change during the 10-year period from 1972 to 1982 [46], a more recent study conducted over a 40-year period from 1974 to 2014 indicated a significant increase in the standard inversion arrangements and a decrease of the 2R1 variant in populations across the *An. messeae* range in Russia; these changes correlate with the increase in year-round temperatures [47].

Novikov and Kabanova introduced the idea that combinations of inversions in natural populations of *An. messeae* represent two distinct chromosomal complexes [48]. The variant X0 is associated with 2R0, 3R0, and 3L0, and, alternatively, inverted X1 and X2 tends to associate with 2R1, 3R1, and 3L1 [48,49,50]. These chromosomal complexes confer differences in female fecundity, viability of imago and larvae, food preferences, rate of development, relationship with predators and parasites, and sensitivity to the toxins of *Bacillus thuringiensis subsp. israelensis* [43,49,51,52,53,54,55]. Later, Novikov referred to these chromosomal complexes as cryptic genetically isolated forms, named “A” and “B” [56] that, however, have overlapping inversion polymorphisms and cannot be distinguished by any fixed inversion differences. Recent ITS2 sequencing studies demonstrated that form “A” of *An. messeae* is synonymous with *An. daciae* [57,58].

In this study, we characterized chromosome and genomic differentiation of *An. messeae* and *An. daciae* in three populations from the Moscow region (Novokosino, Noginsk and Yegoryevsk) in Russia based on ITS2 sequencing and karyotyping of 289 individual mosquitoes. Additionally, whole-genome sequencing of five specimens from each species from one population was also performed to characterize genomic diversity and divergence.

## 2. Materials and Methods

### 2.1. Field Collection and Material Preservation

Three hundred mosquito larvae were collected in the summer of 2016 in three locations of the Moscow region, Russia: Novokosino (55°44′02.6″ N, 37°50′33.5″ E), Kolyshkino boloto in Noginsk (55°53′57.4″ N, 38°26′17.9″ E), and the village of Parykino near Yegoryevsk (55°17′17.9″ N, 39°22′03.5″ E). Positions of the mosquito breeding sites are shown at the map that was developed using OpenStreetMap data [59] (Figure 1). The Novokosino population is located within the boundaries of Moscow. The ponds were characterized by a high oxygen level (1.8–4.0 mg/L) and high density of larvae (93 and 38 per sq. m., respectively) in Novokosino and Noginsk and by a low oxygen level (0.8 mg/L) and low density of larvae (25 per sq. m.) in Yegoryevsk. *Anopheles* larvae were collected by the dipping method and then fixed in Carnoy’s solution (1:3 acetic acid: ethanol). Larvae were dissected and each individual mosquito was numbered (Appendix A). Each larva was divided into two parts: head with thorax and abdomen. These dissected parts were placed into separate tubes for further analysis. Heads with thoraxes were kept in Carnoy’s solution and abdomens were placed in 70% ethanol.

### 2.2. Genotyping

The abdomens from each of the 300 specimens were used for DNA extraction. For individual homogenization, sterile 1.5 mL tubes were used to prevent the risk of contamination. Genomic DNA was extracted from the specimens using a standard protocol for the Qiagen DNeasy Blood and Tissue Kit (Qiagen, Germantown, MD, USA) with slight modifications. Samples were homogenized in 45 µL of extraction buffer with 5 µL added Proteinase K and incubated at 56 °C overnight. DNA elution was performed in 100 µL of water.

ITS2 from rDNA was amplified using the forward universal primer designed by Proft [60] 5′-ATCACTCGGCTCTCGTGGATCG-3′ (Tm = 64.5 °C) and the reverse primer used by Novikov [56]: 5′-ATGCTTAAATTTAGGGGGTA-3′ (Tm = 54.2 °C). This primer combination was chosen based on the longer amplicon length (613 bp) and high rate of PCR success (100%). Hot start ImmoMix^TM^ polymerase reaction mix (Bioline, Taunton, MA, USA) was used for the PCR reaction. The PCR mixture contained 1–2 µL of DNA template (depending on concentration), 1 µL of both forward and reverse primers at 10 µM concentration, and 10 µL of 2× Immomix reaction-mix. Water was added to the mixture to make 20 µL of total volume. Amplification was performed using a thermal cycler (Eppendorf, Hauppauge, NY, USA) with the following programmed parameters: initial denaturation at 95 °C for 10 min, followed by 35 cycles of 95 °C for 15 s, 55 °C for 30 s, and 72 °C for 30 s, and a final extension step at 72 °C for 5 min. The reaction was then placed on hold at 4 °C.

For DNA sequencing, amplicons were visualized using gel electrophoresis and then purified with a Wizard^TM^ PCR Clean Up kit (Promega, Fitchburg, WI, USA). Concentrations of purified PCR products were measured using a Nanodrop (Thermo Fisher Scientific, Haverhill, MA, USA). PCR products were mixed with either forward or reverse primers and sequenced on the Sanger sequencing platform at the Virginia Biocomplexity Institute. The majority of samples were sequenced with both forward and reverse primers to eliminate the possibility of contamination and to confirm the presence of single-nucleotide polymorphisms (SNPs) on both DNA strands. Samples from Noginsk were sequenced with forward primers only.

A total of 289 specimens were successfully genotyped and further analyzed. ITS2 sequences were analyzed using the SeqScape v 2.5 software (Thermo Fisher Scientific, Haverhill, MA, USA). The ITS2 sequence of *An. messeae* (Genbank: AY648982) [32] was used as a reference. Base calling was performed using KB basecaller with mixed base identification enabled. The previously described SNPs in positions 211, 215, 217, 412, and 432 distinguish ITS2 sequences of *An. measseae* from *An. daciae* and the new diagnostic SNP in position 150 were investigated in detail. To verify the appropriateness of the assignment of a nucleotide base in these positions, all diagnostic SNPs were analyzed manually. All ITS2 sequences were submitted to Genbank to obtain accession numbers (Appendix A).

### 2.3. Sequencing and Analysis of Individual Mosquito Genomes

The individual mosquito genomes of *An. daciae* (ME36, ME71, ME72, ME78, ME80) and *An. messeae* (ME3, ME14, ME24, ME25, ME77) were sequenced using Illumina HiSeq 4000 platform in the Fasteris, SA sequencing center. Standard genomic Nano libraries for 2× 150 bp paired-read sequencing were prepared for each sample to obtain 10 million reads aiming at 10× genome coverage. The raw reads were checked with FastQC software [61] and then trimmed using Trimmomatic v 0.36 [62] to remove bases with poor base quality and the remains of adapters with the following flags: LEADING:24 TRAILING:24 SLIDINGWINDOW:4:24 MINLEN:50. The trimmed reads were mapped to the *An. atroparvus* genome AatrE2 [63,64] using the BWA mem algorithm [65] with -M flag and sorted with the *samtools sort* command of the SAMtools package [66]. The single-nucleotide variants (SNVs) were called using *samtools mpileup | bcftools call* command [67] with the following settings: minimum mapping quality = 55 (-q 55), minimum base quality = 26 (-Q 26), calling only SNVs (-I), and multiallelic calling model (-m). The resulting VCF file was filtered out with VCFtools [68] allowing a minimal phred-scaled SNV quality = 400 (--minQ 400), minimal genotype quality = 10 (--minGQ=10), no more than 1 missing genotype per SNV (--max-missing-count 1), minimal distance between the SNVs = 10 Kbp (--thin 10,000) to reduce the number of linked loci, and allow only biallelic SNVs. The scaffold-based coordinates of the SNVs were converted to the chromosome-based coordinates with the Chromosomer tool [69] based on comparison of the cytogenetic maps of *An. measseae* [40] and *An. atroparvus* [70].

We calculated the index of population differentiation, Fst [71] between *An. messeae* and *An. daciae* using the VCFtools function ---weir-fst-pop in the sliding nonoverlapping 5 Kbp windows along the genome. To infer genetic ancestry of the individual genomes, we used ADMIXTURE software [72] to assign fractions of the individual genomes to putative ancestral genetic clusters with K = 2 and only autosomal sequences. Principal Component Analysis (PCA) based on the whole genome sequences was carried out with SNPRelate software [73].

### 2.4. Karyotyping

Salivary glands were dissected from the larval thorax for preparations of polytene chromosomes. Chromosome preparations were made by the standard lacto-aceto-orcein method [74]. Polytene chromosomes were visualized using an Eclipse E200 light microscope (Nikon, BioVitrum, Moscow, Russia). Specimens were karyotyped using a standard chromosome map for the salivary glands of *An. messeae* [40,47,74]. All inversions—X1, X2, X4, 2R1, 3R1, and 3L1—were considered and the karyotype of each specimen was described for the whole chromosomal complement (Appendix A). We attempted to karyotype a total of 300 specimens (100 samples from each location) but only 289 samples were successfully karyotyped.

### 2.5. Statistical Analysis of Chromosomal Inversions

We used the software GENODIVE [75] to statistically analyze the population architecture of the chromosomal polymorphism. We exploited the AMOVA framework [76,77] to test for significance of the deviations from the Hardy-Weinberg equilibrium with 10k permutations and applied the Bonferroni correction for multiple testing. The pairwise Fst values [71,76,77] between the studied populations were calculated based on 10k permutations using only autosomal inversions and the Bonferroni correction was applied. Finally, we performed the Principal Component Analysis with GENODIVE software using the frequencies of autosomal inversions on the population level to infer and visualize the relationship between the species and its populations.

## 3. Results

### 3.1. Molecular Structure of ITS2 in *An. messeae* and *An. daciae* from the Moscow Region

In this study, we identified *An. messeae* and *An. daciae* based on ITS2 sequencing in three populations in Moscow region. As it was reported earlier, the two species differ from each other by five nucleotide substitutions in the ITS2 region of their rDNA sequence [32]. These ITS2 sequences contained the following nucleotides in *An. messeae*: T, T, C, G, G in positions 211, 215, 217, 412 and 432, respectively. Whereas *An. daciae* had A, A, T, A, C in the same positions [32]. Here and below, the numbers of the nucleotide positions are given in correspondence to the original sequence submitted under GenBank number AY648982 [32]. In our samples, all 160 of *An. messeae* had previously described nucleotides (Figure 2A, Table 1). Contrary to most previous descriptions, 119 specimens that we identified as *An. daciae* had ITS2 sequences with heterogeneous substitutions in the first three positions. We found the following double peaks in sequence chromatograms: A+T (W), A+T (W) and T+C (Y) in positions 211, 215, and 217 but single picks A and C in positions 412 and 432 (Figure 2B, Table 1). Similar sequencing results were obtained for *An. messeae* and *An. daciae* from samples in the Novosibirsk region of Russia [58]. In addition to the originally described five substitutions [32], we found another diagnostic nucleotide in position 150 (Figure 2, Table 1). In all our *An. messeae* specimens, we found a double peak A+C (M) in this position, whereas, in the previous report, only nucleotide C was found in this position in *An. messeae* [32]. In all our *An. daciae* specimens, we determined the C nucleotide in position 150 with only two exclusions where a double peak A+C (M) were observed. Based on the sequencing data from 280 samples, we were able to identify only one individual mosquito that we considered a hybrid between *An. daciae* and *An. messeae*, where double peaks were presented in all six diagnostic positions (Figure 2C). Thus, we conclude that unique substitutions in positions 412 and 432—G, G for *An. messeae* and A, C for *An. daciae*—are the most reliable for the species and hybrid diagnostics.

### 3.2. Species Compositions in Moscow Populations

We collected the larvae of *Anopheles* mosquitoes from three locations in the Moscow region (Figure 1). The Novokosino larval breeding site was located within the border of the city of Moscow, the Noginsk breeding site was located 36 km east of Moscow, and the Yegoryevsk breeding site was located in the village of Parykino, which is located 99 km to the southeast of Moscow. All three populations were located near human settlements in the same landscape zone of the Meshcherskaya Lowland territory. The ponds in Novokosino and Noginsk represented typical breeding sites of *Anopheles* mosquitoes (Figure 3A,B). The open water surface was exposed to the sun and the abundance of floating vegetation created favorable conditions for the development of mosquito larvae. On the contrary, the larval pond in Yegoryevsk was shaded by trees and contaminated with tree litter (Figure 3C). The three locations differed in the predominant aquatic and near-water plants.

Species composition in the three locations was determined by sequencing ITS2. We found a higher proportion of *An. messeae* in Novokosino where the percentages of *An. messeae* to *An. daciae* were 73% to 26%, respectively, and in Noginsk where the percentages of *An. messeae* to *An. daciae* were 66.3% and 32.6%, respectively (Figure 1). A different species composition pattern was found in Yegorevsk, with a ratio of 25.5% to 66% of *An. messeae* to *An. daciae*, respectively. One hybrid mosquito (1%) was found in Yegorevsk. In addition, four (4.3%) *An. beklemishevi* and three (3.2%) *An. maculipennis* larvae were identified based on karyotyping [26,78] and ITS2 PCR product length [60] in Yegorevsk and one (1.1%) *An. maculipennis* in Noginsk and Novokosino each.

Thus, our study demonstrated that *An. messeae* dominated in the typical malaria mosquito open, sunny breeding sites of Novokosino and Noginsk, with an increased oxygen content of 1.8–4.0 mg/L (Figure 3A,B). The density of larvae in these breeding sites was 93 and 38 per sq. m., respectively. *An. daciae* had the advantage in the unusually shaded breeding site of Yegoryevsk, with high water saprobity and a low oxygen content of 0.8 mg/L (Figure 3C). The density of larvae in this pool was only 25 per sq. m. It is unclear, if such differentiation is related to the species-specific female oviposition preferences or *An. daciae* larvae being better adapted to tolerate unfavorable developmental conditions.

### 3.3. Chromosomal Inversions in *An. messeae* and *An. daciae*

Five inversions in four chromosomal arms, X1, X4, 2R1, 3R1, and 3L1, have been previously described in *An. messeae* in the Moscow region [30,79,80]. We analyzed inversion polymorphism in 280 specimens from three locations in Moscow region. The most common inversions in *An. messeae* and *An. daciae* polytene chromosomes are shown in Figure 4. Inversion X4 is rare and was only found in the Moscow region at a low frequency. The other inversions are common in the populations of *An. messeae* across the range of its distribution [17]. In our study, each genotyped sample was assigned to a corresponding karyotype (Appendix A). According to the common nomenclature of inversions in *An. messeae*, standard arrangements are referred to as 00 variants, inverted as 11 variants, and heterozygotes as 01 or 14 variants. Inversions on the X chromosome in hemizygous males are referred as 0, 1, and 4 variants.

The frequencies of the chromosomal inversion variants were calculated for both *An. messeae* and *An. daciae* in three populations (Appendix A, Figure 5). Our analysis demonstrated that frequencies of all chromosomal inversions were significantly different between *An. messeae* and *An. daciae*. The most dramatic difference was observed in the X sex chromosome (Figure 5A). Inversion X1 was fixed in *An. messeae*. We found no standard arrangement X0 in all three populations of *An. messeae*. In contrast, this arrangement was present in all three *An. daciae* populations with almost equal frequencies of standard and inverted variants. The endemic inversion X4 was found only in *An. messeae* populations in all three locations, although in low frequencies. The X4 arrangement was observed only in X14 heterozygote females and X4 hemizygote males. Although the inverted autosomal variants 2R1 and 3R1 were present in *An. daciae*, we found significantly higher frequencies of the inverted variants in *An. messeae* populations (Figure 5B–C). Moreover, inverted variant of the 3L1 inversion were only found in *An. messeae* in low frequency (Figure 5D). In addition to the previously described chromosomal inversions, we discovered a new inversion, 2R4 (7A–11A), that was found as a heterozygote 2R04 in *An. daciae* from the Yegoryevsk population (Figure 4B, Figure 5B). This unique inversion was found for the first time in only one location. Interestingly, a hybrid between *An. daciae* and *An. messeae* carried an unusual karyotype of an X04 heterozygote of the standard variant X0, which is typical for *An. daciae* and the inverted variant X4, which was only observed in *An. messeae* (ME21 sample in Appendix A). Overall, our study demonstrated that inversion frequencies were higher on the X chromosome than on the autosomes in *An. daciae*, whereas inversion frequencies were higher on autosomes in *An. messeae*.

Our statistical analysis demonstrated that the frequencies of homozygotes and heterozygotes within *An. messeae* and *An. daciae* samples were in Hardy-Weinberg equilibrium for autosomal inversions but deviated from Hardy-Weinberg equilibrium for sex-linked X-chromosomal inversions in the Noginsk population of *An. daciae* and in the Yegoryevsk population of both species (Table 2). Pairwise analysis of population differentiation revealed no significant population differentiation within the species (Fst = 0.003–0.046 within *An. messeae*, Fst = −0.017–0.01 within *An. daciae*, nonsignificant, Table 3), but highly significant Fst values between the two studied species. The interspecies Fst values ranged from 0.076 to 0.28 and were highly significant for a majority of the pairwise comparisons after the Bonferroni correction for multiple testing was applied. The PCA based on the frequency of autosomal inversions only reliably separated the two species by the PC1, accounting for 96.2% of the total variance (Figure 6).

### 3.4. Genome Sequence Analysis of *An. messeae* and *An. daciae*

To compare genomic differentiation between the species, we sequenced 10 individual mosquito genomes from the Yegoryevsk population: five individuals each of *An. messeae* and *An. daciae*. For each sequenced library, 88.24% to 90.69% of the reads were aligned to the *An. atroparvus* genome as a reference [64]. The average coverage of the sequenced genomes was 20× and 1,858,004 high-quality SNVs were identified in total after the SNV calling and the filtering step. Analysis of the fixation index (Fst) distribution along the 5 kb genomic windows revealed small or modest genetic differentiation between the species on autosomes (mean Fst = 0.027), and dramatic increases in the level of genetic differentiation on the X chromosome (mean Fst = 0.331, Figure 7). A large increase in genomic differentiation is seen in the telomeres, centromeres, and in the middle part of the X chromosome. The increase in genomic differentiation in the middle part of the chromosomes likely overlaps with the location of the highly polymorphic inversion X1 and is caused by the low recombination inside the inversions. Some increases in differentiation between the species was also observed in the centromeres of the autosomes, which are regions of low recombination. PCA based on autosomal polymorphisms reliably separated the two species by the PC1 (Figure 8A), accounting for 14.69% of the total variance. The ADMIXTURE analysis allowed us to separate individuals into two clusters (K = 2) based on their species status and on autosomal SNVs, but it also indicated a considerable level of admixture between the species (Figure 8B). Two out of 10 individual mosquitoes were admixed and may represent second or third generation hybrids between the two species. This result suggests that hybridization between the two species is still ongoing in the populations although a larger sample size is needed for a precise estimation of the current gene flow amount.

## 4. Discussion

### 4.1. Chromosomal Inversions Differentiate *An. messeae* and *An. daciae*

Our study demonstrated chromosomal differentiation between *An. messeae* and *An. daciae* that is more pronounced on the X sex chromosome. Inversion X1 was found only as a fixed arrangement in *An. messeae*, whereas it was polymorphic in *An. daciae*. Another inversion, X4, was only found as a heterozygote or hemizygote states in *An. messeae*. Although the 2R1 and 3R1 autosomal inversions were found in both species, the frequencies of these inversion were much higher in *An. messeae* than in *An. daciae*. Inversion 3L1 was found at low frequency only in *An. messeae*. Rare inversion 2R4 was found as heterozygotes only in *An. daciae*. These data indicate that *An. messeae* is a more polymorphic species at the chromosomal level than *An. daciae*. PCA analyses conducted based on autosomal inversions clearly separated the two species, suggesting the presence of a reproductive barrier between the species.

Chromosomal analysis is considered to be a powerful tool for determining the taxonomic status of a species and for population analyses. Studies conducted in the past using fruit flies and malaria mosquitoes demonstrated that the species often have fixed inversion differences in their chromosomes with rare occurrences of homosequential species [2,11]. For example, fixed chromosomal differences discriminated *An. maculipennis* and *An. messeae* [26], and *An. sacharovi* and *An. martinius* [27]. Another species from the Maculipennis group, *An. beklemishevi*, was identified based on multiple fixed chromosome differences [28,29]. Based on the banding pattern of polytene chromosomes, species in the *An. gambiae* complex have been differentiated by 10 fixed inversions [11]. Five of these inversions are located on the X chromosome separating species into three groups possessing the compound Xag inversion, the standard X arrangement, and the compound Xbcd inversion [81]. Intensive studies of inversion polymorphism within the major malaria vector *An. gambiae* identified five chromosomal forms: Bamako, Savanna, Mopti, Forest, and Bissau [82,83]. Interestingly, some of the combinations of chromosomal inversions have never been observed in nature, suggesting the presence of reproductive barriers within the populations. One nucleotide difference in ITS2 led to the discovery of the M and S forms of *An. gambiae*, which have now been elevated to species status as *An. gambiae* (former S form) and *An. coluzzii* (former M form) [84]. Originally, it was shown that the M and S molecular forms of *An. gambiae* correlate with the previously identified chromosomal forms, Mopti and Savanna. However, later studies determined that these inversion patterns did not always correspond to *An. gambiae* and *An. coluzzi* across the African continent [85,86]. Interestingly, all chromosomal inversions located in the X sex chromosome in the *An. gambiae* complex are fixed between the species, suggesting that this chromosome may play an important role in speciation. In contrast, 2 polymorphic inversions were found on the X chromosome in *An. daciae* and *An. messeae* in our study.

### 4.2. A Whole-Genome Analysis Determines the Genome-Wide Divergence between *An. messeae* and *An. daciae*

In addition to the cytogenetic analysis, we performed whole-genome sequencing of five specimens of *An. messeae* and *An. daciae* from the Yegoryevsk population. Our study identified the whole-genome divergence between the two species. Genomic divergence was most pronounced on the X chromosome and in the pericentromeric areas of the autosomes, suggesting a role of the X chromosome and heterochromatin in species differentiation. At the same time, the level of genomic divergence along the autosomal chromosomal arms was much lower than in chromosome X indicating that the gene flow is still ongoing in most of the genome or a high level of ancestral shared polymorphism is still present. Although PCA and ADMIXTURE analyses based on autosomal SNVs clearly separated the two species into two clusters, we identified two admixed individuals in the sample of 10. Together with a single first-generation hybrid identified by ITS2 sequencing, we can conclude that hybridization occurs between *An. messeae* and *An. daciae* in the Moscow region. Further genomic studies from different geographic locations are required to estimate hybridization frequencies and the level of genomic differentiation between the two species.

Similar genomic divergence was discovered between *An. gambiae* and *An. coluzzi*, formerly the molecular forms S and M, respectively, of the *An. gambiae* complex [87]. As we mentioned above, sequencing of the ITS2 in *An. gambiae* revealed an intriguing one nucleotide difference in the natural populations suggesting incipient speciation within *An. gambiae* [87,88]. Despite the ongoing gene flow between the M and S forms [13], the whole-genome comparison based on Affymetrix GeneChip microarrays identified so called “speciation islands” [89], which reside in heterochromatic regions [90] of the genome *of An. gambiae*. Further whole-genome sequencing of the two forms demonstrated a high level of genomic differentiation across the entire genome [91], which was especially high near the centromeres and inside the polymorphic inversions. As a result of these studies, the M and S forms were elevated to species status and named *An. coluzzi* and *An. gambiae*, respectively [84]. Interestingly, the level of the genomic divergence between *An. gambiae* and *An. coluzzii* was slightly higher on chromosome X but not as dramatic as between *An. messeae* and *An. daciae.* Such a high level of the genomic divergence on chromosome X could be explained by the inversions located on this chromosome.

The role of the X chromosome in speciation has been shown for various organisms [92]. The large X-effect and Haldane’s rule are known as two rules of speciation. It has been noticed that the X chromosome has a disproportionally larger effect of on hybrid malfunctions, compared with autosomes [93]. Several theories have been proposed to explain these empirical phenomena. The dominance theory suggests that if alleles driving hybrid dysfunction are recessive and located on the X chromosome, they will be exposed in males mainly affecting the heterogametic sex [94]. Such dominant effects will lead to X-linked genes evolving faster than autosome-linked genes according to the faster-X theory [95]. Alternatively, incompatibility between X and Y chromosomes could be due to a mutual imbalance between meiotic drive genes, which are more likely to evolve on sex chromosomes than autosomes [96].

### 4.3. *An. messeae* and *An. daciae* Differ in Their Ecological Preferences and Behavior

Understanding species composition and ecological preferences is important for the development and application of any strategy for vector control [16]. The malaria mosquitoes *An. daciae* and *An. messeae* occupy regions with temperate oceanic and humid continental climates according to the Eurasian Koppen-Geiger climate classification system [97]. These two species often occur in sympatry [33,35,36,37,38,39,98,99] but their ecological niches do not completely overlap. A recent study conducted in Germany demonstrated that the probability of occurrence of *An. messeae* had a negative correlation with the maximum temperature of the warmest month and altitude, whereas the probability of occurrence of *An. daciae* had a positive correlation [98]. In addition, *An. messeae* negatively correlated with the proportion of variable agricultural land cover and *An. daciae* was negatively correlated with pasture land cover. As a result, *An. messeae* demonstrates a decrease in the probability of occurrence from the north to the south of Germany. In contrast, *An. daciae* has a low probability of occurrence in northwestern and southern Germany. The latter species is also absent in coastal areas of Germany, where *An. messeae* occurs [35]. These two species show differences in seasonal dynamics as well: *An. messeae* is more abundant early and late in the season whereas *An. daciae* is prevalent in the warmest mid-summer months [35,39,58]. Both species have differences in blood feeding behavior: females of *An. messeae* prefer feeding on animals whereas females of *An. daciae* have opportunistic blood choice behavior and feed on birds, humans, deer, and livestock [36]. This finding suggests that *An. daciae* may serve as a bridge vector for arboviruses to humans. Moreover, a female of *An. daciae* infected with the nematode *Dirofilaria repens* was detected in Germany [34].

Our study provides further insights into possible ecological differentiation of the two species with respect to their preference for larval breeding places. Previous studies demonstrated that *An. messeae* prefers large clean water reservoirs with a relatively low dissolved ion content [33,39]. In our study, *An. messeae* also dominated in the typical oxygen-rich (1.8–4.0 mg/L) anophelogenic breeding sites of Novokosino and Noginsk, with the density of *Anopheles* larvae being 93 and 38 per sq. m., respectively (Figure 3A,B). *An. daciae* had an advantage in the atypical breeding place in Yegoryevsk, with high water saprobity, low oxygen content (0.8 mg/L), and low density of *Anopheles* larvae (25 per sq. m.). We conclude that the survival optimum for both species, *An. daciae* and *An. messeae*, is in the zone of temperate deciduous and mixed forests. There is a sufficient number of favorable biotopes with abundant aquatic vegetation for joint development of both species in the temperate zone. Ecological specialization of sibling species becomes obvious with deviations from the optimum, up to the competitive exclusion of one of the species. Such competitive displacement can be observed at the edges of the ranges, in the steppe and southern taiga zones. However, it is obvious that ecological preferences and behavioral differences of *An. messeae* and *An. daciae* and the details of their geographical distribution in Russia and other Eurasian countries requires further investigation.

## 5. Conclusions

A cytogenetic analysis conducted here demonstrated strong differentiation between *An. messeae* and *An. daciae* by polymorphic inversions especially on the X chromosome. Inversion X1 was completely fixed in all three populations of *An. messeae*, whereas frequencies of the inverted and standard arrangements in *An. daciae* were almost equal in all populations. Although the two most abundant polymorphic autosomal inversions were found in both species, their frequencies were much higher in *An. messeae* than in *An. daciae*. The genomic analyses of 10 individual mosquitoes demonstrated a genome-wide differentiation along the chromosomes. The most dramatic genomic differences were found on the inversion-rich sex chromosome X, suggesting a role of this chromosome in reproductive isolation between the species. The ADMIXTURE cluster analysis of 10 individual genomes identified two clusters corresponding to *An. messeae* and *An. daciae*. The presence of two admixed individuals suggests that hybridization is ongoing between the populations of *An. messeae* and *An. daciae*. Overall, our study suggests that chromosome inversions may play important role in diversification of these two species. We demonstrated that *An. messeae* and *An. daciae* represent closely related cryptic species with incomplete reproductive isolation that are able to maintain their genome integrity in sympatry despite genetic introgression.

## Figures and Tables

**Figure 1 genes-11-00165-f001:**
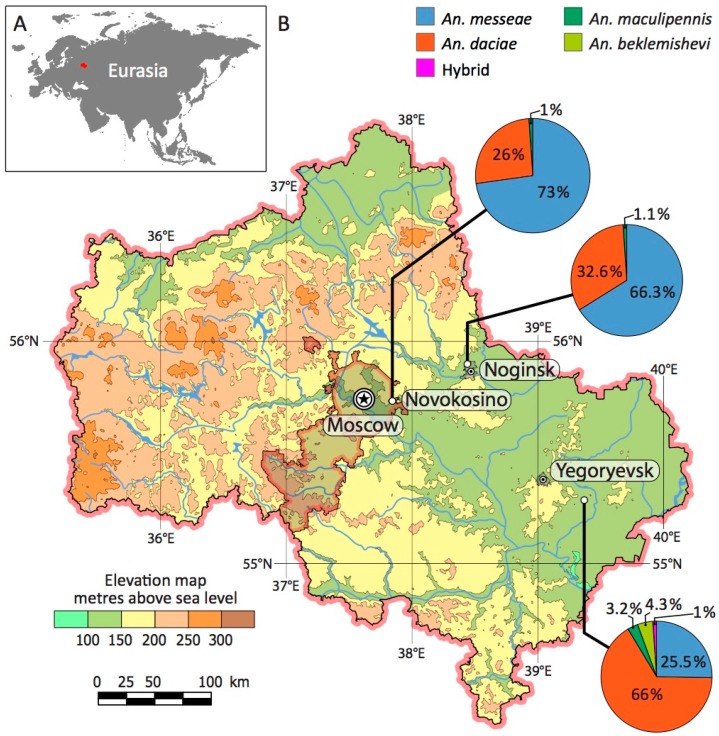
Location of the Moscow region in Eurasia (**A**) and collection sites of mosquito larvae (**B**). The ratios of *An. messeae*, *An. daciae*, their hybrids, *An. maculipennis*, and *An. beklemishevi* are shown as pie charts for each population. The charts show different proportions of the species in three compared populations. The map was developed using OpenStreetMap data [59].

**Figure 2 genes-11-00165-f002:**
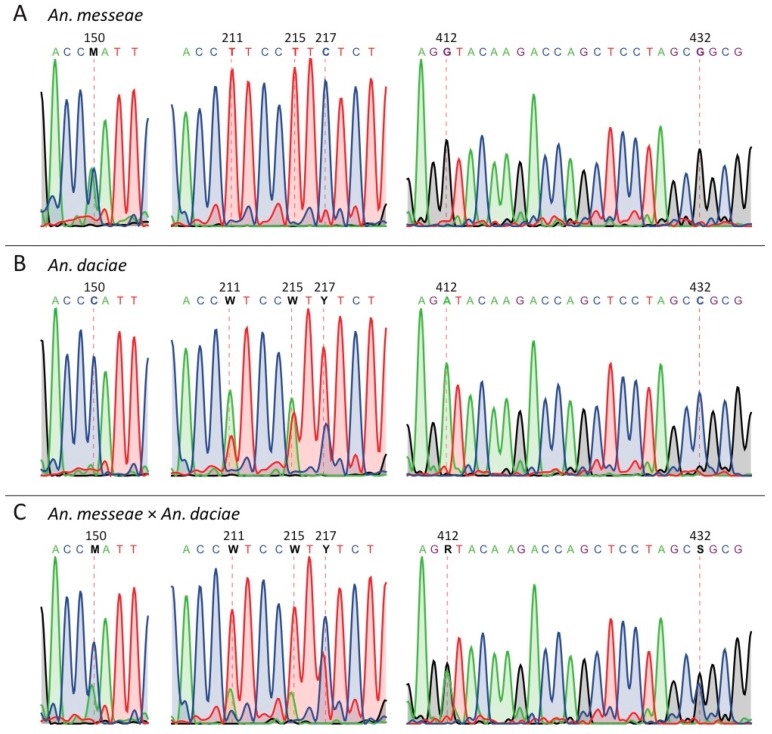
Examples of ITS2 sequence chromatograms for *An. messeae* (**A**), *An. daciae* (**B**), and their hybrid (**C**). Dash lines indicate positions of SNPs that distinguish the two species. Chromatograms indicate the presence of double picks in position 150 of *An. messeae* and in positions 211, 215, and 217 of *An. daciae*. The specimen with double picks in position 150, 211, 215, 217, 412, and 432 was identified as a hybrid between *An. messeae* and *An. daciae*.

**Figure 3 genes-11-00165-f003:**
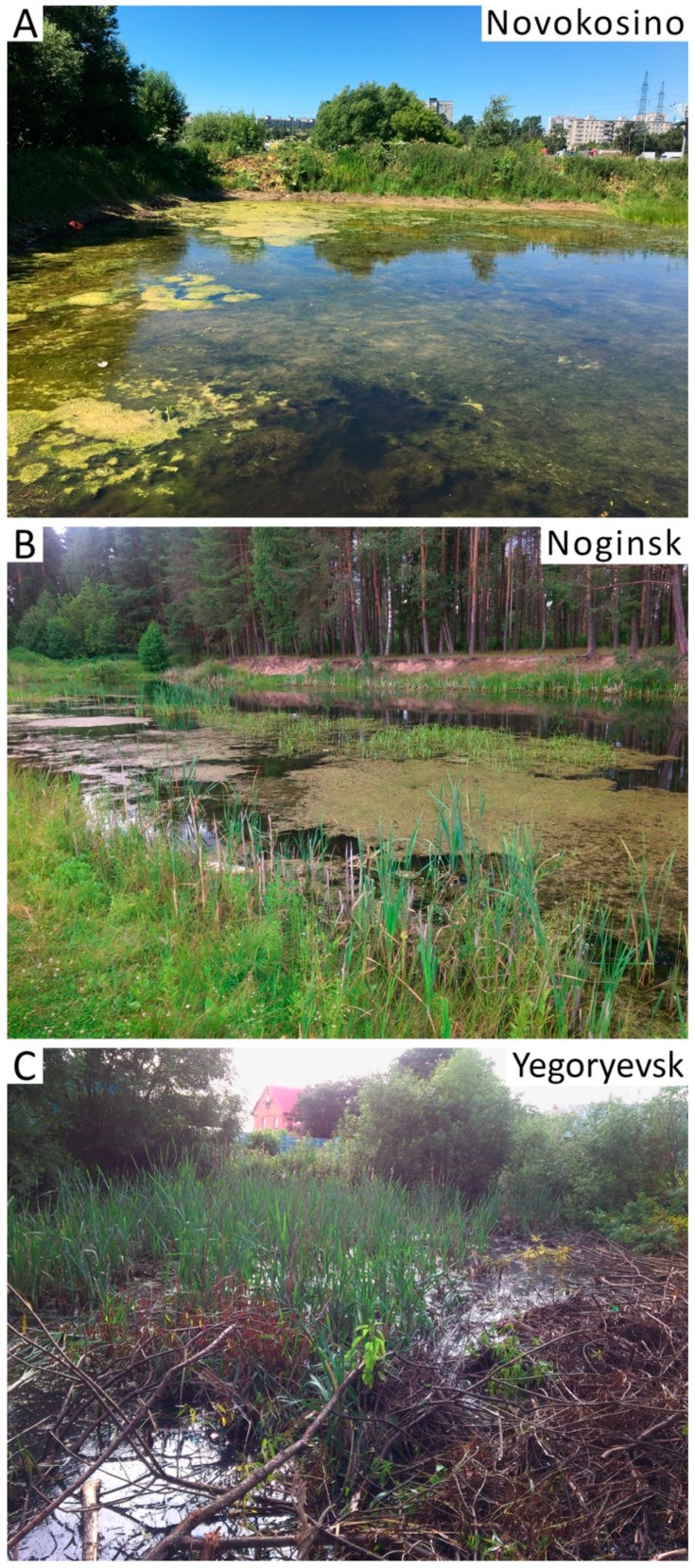
Mosquito breeding sites in Novoskosino (**A**), Noginsk (**B**), and Yegoryevsk (**C**). Ponds in Novokosino and Noginsk are preferred by *An. messeae* and represent typical for *Anopheles* sunny larval breeding sites with open water and abundant vegetation. The water reservoir in Yegorevsk, preferred by *An. daciae*, is shaded and characterized by high water saprobity.

**Figure 4 genes-11-00165-f004:**
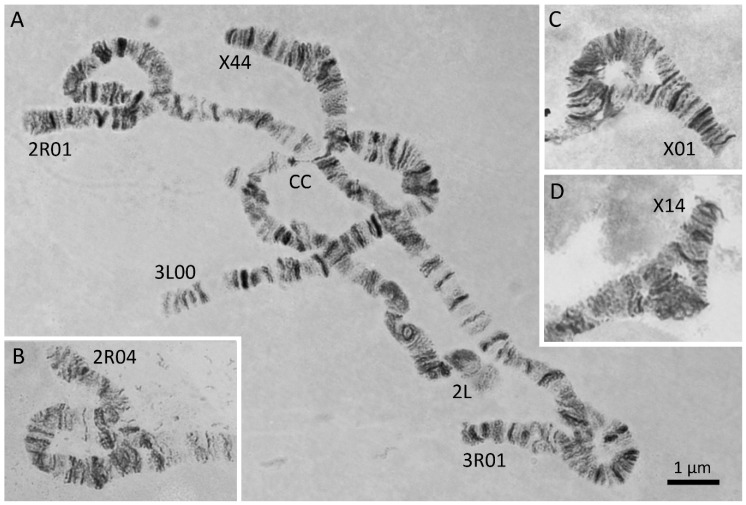
Inversions in polytene chromosomes of *An. messeae* and *An. daciae*. The specimens in the Moscow region are characterized by the presence of 4 highly polymorphic inversions X1, 2R1, 3R1, and 3L1, and 2 rare endemic inversions X4 and 2R4. A rare karyotype X44, 2R01, 3R01, and 3L00 in *An. messeae* is shown on panel **A**. Another rare inversion heterozygote, 2R04, in *An. daciae* is shown on panel **B**. Inversion heterozygotes X01 in *An. daciae* and X14 in *An. messeae* are indicated on panels **C** and **D**, respectively. CC stands for chromocenter.

**Figure 5 genes-11-00165-f005:**
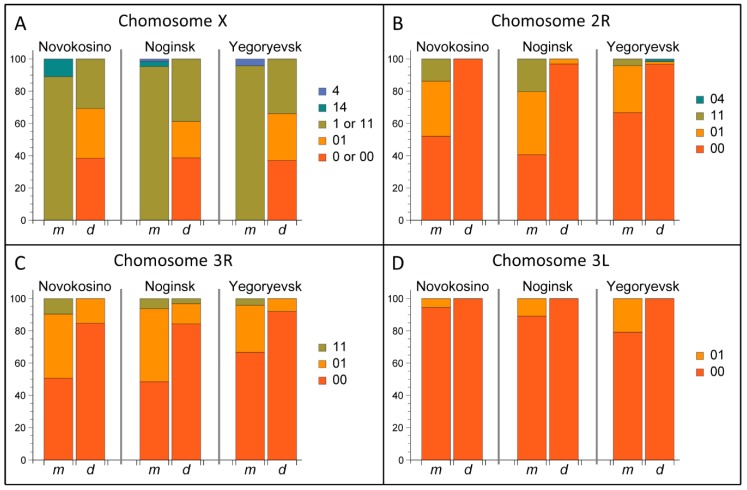
Inversion frequencies in *An. messeae* and *An. daciae* from the Novokosino, Noginsk, and Yegoryevsk populations. Frequencies of inversions: X0, X1, and X4 (**A**); 2R1 and 2R4 (**B**); 3R1 (**C**); and 3L1 (**D**) are shown by charts. Proportions of standard, inverted, and heterozygote arrangements are shown by different colors. Chromosome X is almost monomorphic in *An. messeae* but is highly polymorphic in *An. daciae* in the three Moscow populations. Although all autosomal inversions are present in both species, polymorphism is higher in *An. messeae* than in *An. daciae*. Rare inversions, X4 and 2R4, were found in low frequencies in *An. messeae* and *An. daciae*, respectively.

**Figure 6 genes-11-00165-f006:**
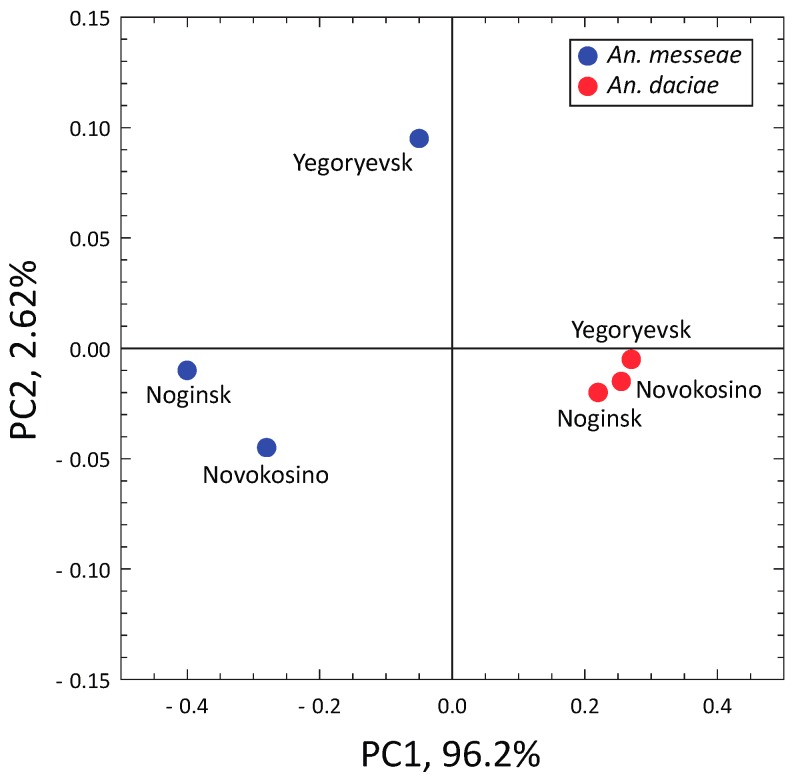
Interpopulation PCA plot based on the frequencies of the autosomal chromosomal inversions in three populations of *An. messeae* and *An. daciae*. Species are indicated by different colors. PCA analysis separates *An. messeae* and *An. daciae* along the PC1.

**Figure 7 genes-11-00165-f007:**
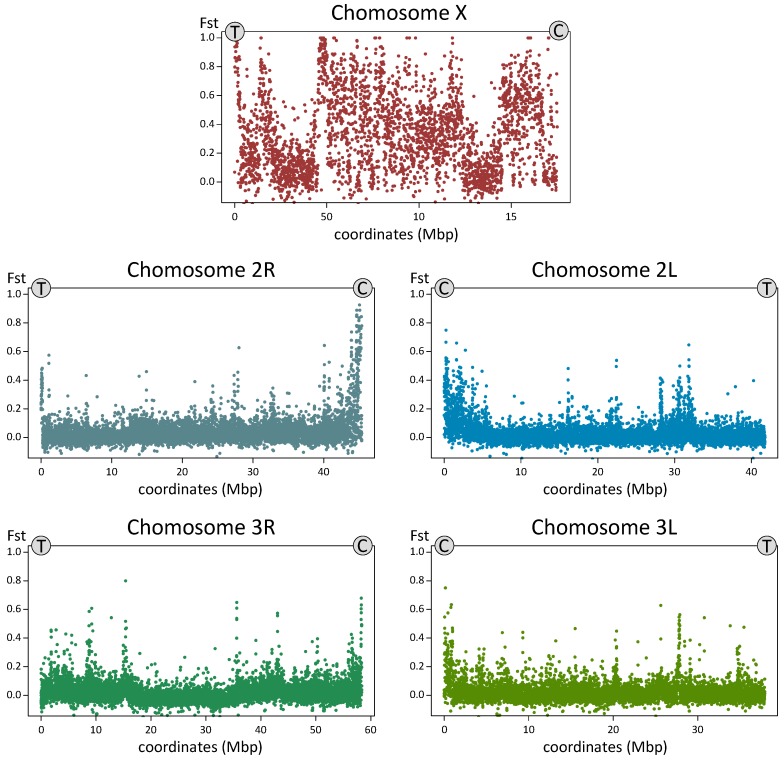
Level of genetic differentiation (Fst) between *An. messeae* (n = 5) and *An. daciae* (n = 5) along the chromosomal arms. Each dot represents a 5 kb window. The y-axis represents Fst values and the x-axis represents the genomic coordinates (Mbp). The X chromosome demonstrates the highest Fst values while the autosomal arms have a low overall level of differentiation, which is elevated in the centromeric regions. T and C stands for telomeres and centromeres, respectively.

**Figure 8 genes-11-00165-f008:**
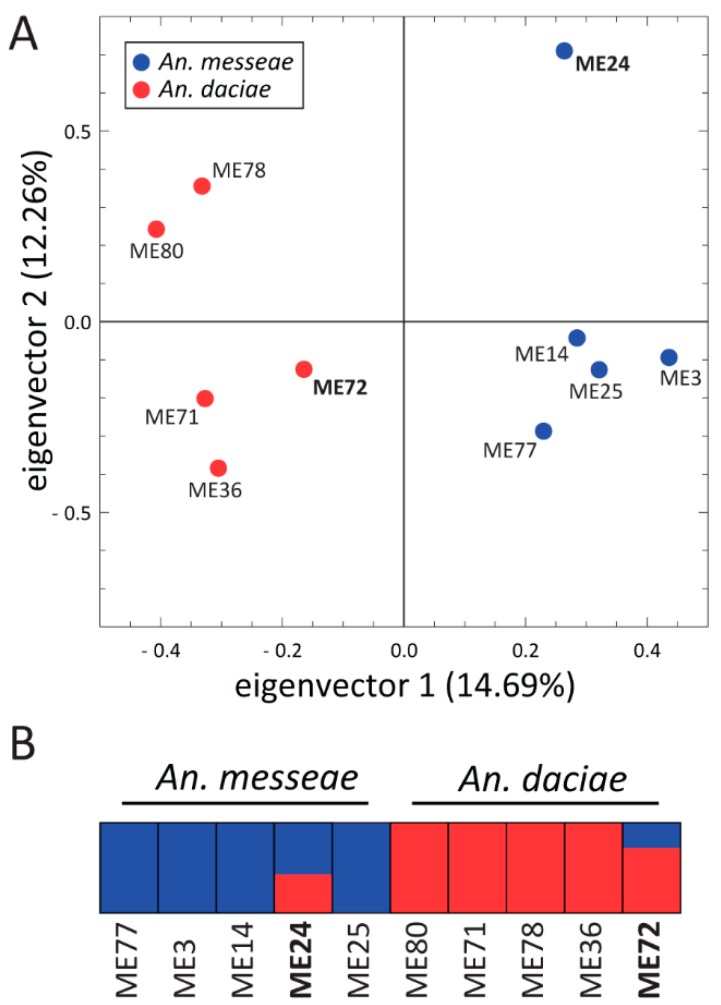
Principal Component Analysis (**A**) and ADMIXTURE (**B**) plots based on the autosomal SNVs of the whole genomes of *An. messeae* and *An. daciae*. Species are shown by different colors. PCA (**A**) reliably differentiates two species by the PC1. Each bar (**B**) represents the proportion of ancestral species in a given individual. Two admixed individuals (shown in bold) were identified.

**Table 1 genes-11-00165-t001:** Variations in ITS2 genotypes of *An. messeae* and *An. daciae* from three populations in the Moscow region.

Species	Genotype						Population		
	150	211	215	217	412	432	Novokosino	Noginsk	Yegoryevsk
*An. daciae*	C	W(A+T)	W(A+T)	Y(C+T)	A	C	24	31	60
	C	W	W	T	A	C	2	0	0
	M(A+C)	W	W	Y	A	C	0	0	2
*An. messeae*	M(A+C)	T	T	C	G	G	73	63	24
Hybrid	M	W	W	Y	R(A+G)	S(C+G)	0	0	1
Total							99	94	87

**Table 2 genes-11-00165-t002:** Statistical significance (*p*-values) of the deviations from Hardy-Weinberg equilibrium in the studied populations by inversion polymorphism. *—results are significant with α < 0.05 after applied Bonferroni correction.

Population, Species	X	2R	3R	3L
Novokosino, *An. messeae*	0.817	0.071	0.433	0.958
Novokosino, *An. daciae*	0.046	Monomorphic	0.878	Monomorphic
Noginsk, *An. messeae*	0.051	0.111	0.302	0.838
Noginsk, *An. daciae*	0.002 *	1.000	0.242	Monomorphic
Yegoryevsk, *An. messeae*	0.022 *	0.595	0.601	0.791
Yegoryevsk, *An. daciae*	0.001 *	0.992	0.922	Monomorphic

**Table 3 genes-11-00165-t003:** Pairwise Fst values between the studied populations of *An. messeae* and *An. daciae* based on the frequency of autosomal arrangements. *—results are significant with α < 0.05 after applied Bonferroni correction.

Population Species	Novokosino *An. messeae*	Novokosino *An. daciae*	Noginsk *An. messeae*	Noginsk *An. daciae*	Yegoryevsk *An. messeae*	Yegoryevsk *An. daciae*
Novokosino *An. messeae*	--	0.159 *	0.003	0.143 *	0.021	0.218 *
Novokosino *An. daciae*	0.159 *	--	0.211 *	−0.017	0.095 *	0.000
Noginsk *An. messeae*	0.003	0.211 *	--	0.197 *	0.046	0.280 *
Noginsk *An. daciae*	0.143 *	−0.017	0.197 *	--	0.076	0.010
Yegoryevsk *An. messeae*	0.021	0.095 *	0.046	0.076	--	0.164 *
Yegoryevsk *An. daciae*	0.218 *	0.000	0.280 *	0.010	0.164 *	--

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
