# Peer review of "Chromosome and Genome Divergence between the Cryptic Eurasian Malaria Vector-Species Anopheles messeae and Anopheles daciae"

_genes, 2020, doi:10.3390/genes11020165_

Round 1
Reviewer 1 Report
The manuscript by Naumenko et al. “Chromosome and genome divergence between the cryptic Eurasian malaria vector-species Anopheles messeae and Anopheles daciae” analyses the natural populations of Anopheles mosquitoes from three localities in the Moscow region where coexist these two cryptic species. In addition to determine the species composition of each locality, this study analyze differences in the chromosomal polymorphism by inversion and the genome divergence between these two species. Both kinds of differences are of great interest to understand speciation when considering pairs of species so closely related as the two here considered.
Comments to the authors:
In general, the manuscript is clearly written but there are some mistakes that must be corrected. Some of them are indicated below as “minor concerns and typos”. In addition, I think that the verb “demonstrate” is overused. I feel that this verb implies that something that had been hypothesized before (or something about which there were contradictory ideas) was finally made clear. Thus, in many of the cases, it should be changed by “show”, “indicate”, “reveal” or “imply”.
While I think that the Figures are all clear and relevant, I would like some changes in Tables. I would appreciate a Table with the frequencies of the chromosomal arrangement. Check carefully numbers in Table 1 because they are in conflict with data in Table S1. According to data in Table S1, there are a total of 99 mosquitoes from Novokosino (24+2+0+73+0 = 99) and 94 from Noginsk (31+0+0+63+0 = 94). In Table 3, data on upper and lower part of the matrix are the same. One of these parts could show the P
I think that the section 3.4 (Species compositions in Moscow populations) should be placed earlier in the Results. Probably just after section 3.1 (Molecular structure of ITS2 in messeae and An. daciae from the Moscow region) which develops the tools that allow to determine the species to which each individual belongs. I consider that this would be a more logical location within the manuscript and could also help to solve the problem that I observed throughout the manuscript where a total of 289 individuals are said to have been analyzed, but table S1 only shows data for 280 (160 A. messeae, 119 A. daciae and 1 hybrid individual). Maybe, the 9 lost individuals correspond to the An. beklemishevi and An. maculipennis reported in lines 356-357 but this is not figured out until the end of results.
As a curiosity matter: why did you name the new inversion as 2R4*? It seems quite different from the classical nomenclature (without *). On the other hand, could you describe better this new inversion indicating the bands affected by their breakpoints? This could be useful for other researchers.
Minor concerns and/or typos
L 22. Use italics “An. messeae”
L 24. Change “However, an inversion polymorphism” to “However, the inversion polymorphism”
L 25. Change “between the species” to “between these species”
L 39. Change “When chromosomal inversion” to “When a chromosomal inversion”
L 44. When say “The role of the chromosomal inversions in adaptation and evolution of different Drosophila species has been studied for several decades [3].” I think that some more references for Drosophila should be given. Certainly, Krimbas & Powell (1992) collects much of the work, but since 1992 drosophilists have also been working on this. For example:
Orengo, Dorcas J; Puerma, Eva; Aguadé, Montserrat. Monitoring chromosomal polymorphism in Drosophila subobscura over 40 years. Entomological Science 19:215-221. Kapun, Martin; Fabian, Daniel K; Goudet, Jérôme; Flatt, Thomas. 2016. Genomic evidence for adaptive inversion clines in Drosophila melanogaster. Molecular Biology and Evolution 33: 1317-1336 Fuller, Zachary L.; Haynes, Gwilym; Richards, Stephen; Schaeffer, Stephen W. 2017. Genomics of natural populations: evolutionary forces that establish and maintain gene arrangements in Drosophila pseudoobscura. Molecular Ecology 26:6539-6562.
L 75. Change “longitude cline” to “latitude cline”
L 78. Change “West-Esat latitude cline” to “West-Est longitude cline”
L 90, 96, 128 and 129. Remove the initials corresponding to the first names of the authors mentioned.
L 97. “However, according to this study,” It is not clear if authors refer to the previous mentioned work [54] or their own present results. I think that it should be “according to the present study”. Otherwise I do not understand how Novikov could define the forms “A” and “B”, but it results confusing.
L 113. Change “an increased” to “a high”
L 164. Check the numbers of the individual mosquitoes used to obtain genomes. In this line appear ME89 but in the Table S1 is indicated ME80. At least, one of these two must be corrected.
L 171. “trimmed reads were aligned to” better “trimmed reads were mapped to”.
L 214. Change “specimen” to “samples”
L 215 – 228. Please rewrite these sentences. It seems odd that the authors say that their results are in opposition to the previous observation, but then explain that there exists a previous work with similar results.
Figure 3. L 255. Change “2R04*, in An. daciae is demonstrated on panels B.” to “2R04*, in An. daciae is shown on panel B.”
Figure 3. on panel A, the meaning of CC should be explained. I understand it corresponds to centromeres?
L 260. “Inversion X1 was fixed in An. messeae” I do not think so. Table S1 shows 12 An. messeae mosquitoes carrying the arrangement X4. I think that this sentence can be deleted because the following sentences explain better what the authors found.
L 262. “this inversion was highly polymorphic in all three An. daciae populations …“ The chromosome is polymorphic, not the inversion (furthermore, in this case should say arrangement because X0 is the standard not inverted chromosome).
L 267-268. “Moreover, inverted variants of the 3L1 inversion were only found in An. messeae in low frequency “. Delete “inverted variants of” I understood that each inversion reported in these species (X1, X4, 2R1, 2R4* 3R1) is unique, i.e. they do not have several variants at the cytological level.
L 280-281. The same problem than before in L260 and L262. Change “Inversion X1 is fixed in An. messeae but is highly polymorphic in An. daciae in the three Moscow populations.“ to “Chromosome X is almost monomorphic in in An. messeae but is highly polymorphic in An. daciae in the three Moscow populations”.
L 290. “(Fst = 0.003 – 0.076 within An. messeae,“ According to data on Table 3, this range of Fst should be 0.003 – 0.046.
L 302. Change “inversions“ to “arrangements
L 308. Use lower case in “20X”
L 313. Change “The large increase in genomic differentiation“ to “A large increase in genomic differentiation”
L 328. Change “The x-axis represents Fst“ to “The y-axis represents Fst”
L 329. Change “the y-axis represents the genomic coordinates“ to “the x-axis represents the genomic coordinates”
First paragraph of discussion should be rewritten to correct the following sentences:
“Inversion X1 was found only as a fixed arrangement in messeae, whereas it was polymorphic in An. daciae.“On the one hand, it is the chromosome that is or not is polymorphic, not a particular inversion. On the other hand, in the three localities An. messeae also shows the arrangement X4 in a non-negligible proportion ranging 2.6 - 6.4% (according to data in Tabla S1).
“Another inversion, X4, was only found as a heterozygote in messeae.“ The expression “a heterozygote” seems to mean that it was only one mosquito with this karyotype in the samples, but Table S1 reveals that there were 11 mosquitoes carrying this arrangement. In addition, not all of them were heterozygotes: there are 9 females (heterozygotes) X14 and 2 males (hemizygotes) X4.
L 449. Use italics “An. messeae”
L 449. Change “Two species demonstrate differences “ to “These two species show differences”
References should be carefully checked. The format of many citations must be corrected to standardize them. Many of the references from 28 to 77 have the title within square brackets [ ].
Supplementary table S1.
“Letters M, W, and Y in genotypes indicate the presence of double picks in ITS2 sequences as follows: M – A and C, W – A and T, Y – T and C.” Could be changed by something like this “The presence of double picks in ITS2 sequences is indicated by the IUPAC nucleotide ambiguity code”. I do this comment because there are two more letters in the table (for the hybrid individual ME21; R – A and G, S – C and G). “Males can be recognized by a single inversion number in the X chromosome karyotype that corresponds to one homolog of this chromosome in males.” Here “inversion” should be changed to “arrangement” since 0 corresponds to the chromosome without inversions.Author Response
Reviewer comments and our responses
Comment 1
In general, the manuscript is clearly written but there are some mistakes that must be corrected. Some of them are indicated below as “minor concerns and typos”. In addition, I think that the verb “demonstrate” is overused. I feel that this verb implies that something that had been hypothesized before (or something about which there were contradictory ideas) was finally made clear. Thus, in many of the cases, it should be changed by “show”, “indicate”, “reveal” or “imply”.
Response
We changed the wording according to the reviewer suggestion.
Comment 2
While I think that the Figures are all clear and relevant, I would like some changes in Tables. I would appreciate a Table with the frequencies of the chromosomal arrangement. Check carefully numbers in Table 1 because they are in conflict with data in Table S1. According to data in Table S1, there are a total of 99 mosquitoes from Novokosino (24+2+0+73+0 = 99) and 94 from Noginsk (31+0+0+63+0 = 94).
Response
We corrected the numbers in the tables.
Comment 3
I think that the section 3.4 (Species compositions in Moscow populations) should be placed earlier in the Results. Probably just after section 3.1 (Molecular structure of ITS2 in messeae and An. daciae from the Moscow region) which develops the tools that allow to determine the species to which each individual belongs. I consider that this would be a more logical location within the manuscript and could also help to solve the problem that I observed throughout the manuscript where a total of 289 individuals are said to have been analyzed, but table S1 only shows data for 280 (160 A. messeae, 119 A. daciae and 1 hybrid individual). Maybe, the 9 lost individuals correspond to the An. beklemishevi and An. maculipennis reported in lines 356-357 but this is not figured out until the end of results.
Response
We agree with the reviewer regarding the order of the subsections in the Results section. We have moved section 3.4 (Species compositions in Moscow populations) right after the section 3.1.
Comment 4
As a curiosity matter: why did you name the new inversion as 2R4*? It seems quite different from the classical nomenclature (without *). On the other hand, could you describe better this new inversion indicating the bands affected by their breakpoints? This could be useful for other researchers.
Response
We changed the name of a new inversion 2R4* to 2R4 to make it consistent with the name for other inversions in An. messeae.
Comment 5
L 22. Use italics “An. messeae”
Response
We changed this.
Comment 6
L 24. Change “However, an inversion polymorphism” to “However, the inversion polymorphism”
Response
We changed this.
Comment 7
L 25. Change “between the species” to “between these species”
Response
We changed this.
Comment 8
L 39. Change “When chromosomal inversion” to “When a chromosomal inversion”
Response
We changed this.
Comment 9
L 44. When say “The role of the chromosomal inversions in adaptation and evolution of different Drosophila species has been studied for several decades [3].” I think that some more references for Drosophila should be given. Certainly, Krimbas & Powell (1992) collects much of the work, but since 1992 drosophilists have also been working on this. For example:
Orengo, Dorcas J; Puerma, Eva; Aguadé, Montserrat. Monitoring chromosomal polymorphism in Drosophila subobscura over 40 years. Entomological Science 19:215-221. Kapun, Martin; Fabian, Daniel K; Goudet, Jérôme; Flatt, Thomas. 2016. Genomic evidence for adaptive inversion clines in Drosophila melanogaster. Molecular Biology and Evolution 33: 1317-1336 Fuller, Zachary L.; Haynes, Gwilym; Richards, Stephen; Schaeffer, Stephen W. 2017. Genomics of natural populations: evolutionary forces that establish and maintain gene arrangements in Drosophila pseudoobscura. Molecular Ecology 26:6539-6562.
Response
We added these citations to the manuscript.
Comment 10
L 75. Change “longitude cline” to “latitude cline”
Response
We changed this.
Comment 11
L 78. Change “West-Esat latitude cline” to “West-Est longitude cline”
Response
We changed this.
Comment 12
L 90, 96, 128 and 129. Remove the initials corresponding to the first names of the authors mentioned.
Response
We changed this.
Comment 13
L 97. “However, according to this study,” It is not clear if authors refer to the previous mentioned work [54] or their own present results. I think that it should be “according to the present study”. Otherwise I do not understand how Novikov could define the forms “A” and “B”, but it results confusing.
Response
We corrected this sentences as “Later, Novikov referred to these chromosomal complexes as cryptic genetically isolated forms, named “A” and “B” [54] that have overlapping inversion polymorphisms and cannot be distinguished by any fixed inversion differences.”
Comment 14
L 113. Change “an increased” to “a high”
Response
We changed this.
Comment 15
L 164. Check the numbers of the individual mosquitoes used to obtain genomes. In this line appear ME89 but in the Table S1 is indicated ME80. At least, one of these two must be corrected.
Response
We changed this.
Comment 16
L 171. “trimmed reads were aligned to” better “trimmed reads were mapped to”.
Response
We changed this.
Comment 17
L 214. Change “specimen” to “samples”
Response
We changed this.
Comment 18
L 215 – 228. Please rewrite these sentences. It seems odd that the authors say that their results are in opposition to the previous observation, but then explain that there exists a previous work with similar results.
Response
We re-wrote the sentence as “Contrary to most previous descriptions, 119 specimens that we identified as An. daciae had ITS2 sequences with heterogeneous substitutions in the first three positions.”
Comment 19
Figure 3. L 255. Change “2R04*, in An. daciae is demonstrated on panels B.” to “2R04*, in An. daciae is shown on panel B.”
Response
We changed this.
Comment 20
Figure 3. on panel A, the meaning of CC should be explained. I understand it corresponds to centromeres?
Response
We wrote that “CC stands for chromocenter.”
Comment 21
L 260. “Inversion X1 was fixed in An. messeae” I do not think so. Table S1 shows 12 An. messeae mosquitoes carrying the arrangement X4. I think that this sentence can be deleted because the following sentences explain better what the authors found.
Response
The reviewer is correct regarding X4 inversion that is polymorphic in An. messeae. However, inversion X1 (that is different from X4) is fixed in An. messeae.
Comment 22
L 262. “this inversion was highly polymorphic in all three An. daciae populations …“ The chromosome is polymorphic, not the inversion (furthermore, in this case should say arrangement because X0 is the standard not inverted chromosome).
Response
We change the sentence as “this arrangement was present in all three An. daciae populations with almost equal frequencies of standard and inverted variants.”
Comment 23
L 267-268. “Moreover, inverted variants of the 3L1 inversion were only found in An. messeae in low frequency “. Delete “inverted variants of” I understood that each inversion reported in these species (X1, X4, 2R1, 2R4* 3R1) is unique, i.e. they do not have several variants at the cytological level.
Response
We changed this sentence to: “Moreover, inverted variant of the 3L1 inversion were only found in An. messeae in low frequency”
Comment 24
L 280-281. The same problem than before in L260 and L262. Change “Inversion X1 is fixed in An. messeae but is highly polymorphic in An. daciae in the three Moscow populations.“ to “Chromosome X is almost monomorphic in in An. messeae but is highly polymorphic in An. daciae in the three Moscow populations”.
Response
We changed this sentence to: “Chromosome X is almost monomorphic in An. messeae but is highly polymorphic in An. daciae in the three Moscow populations.”
Comment 25
L 290. “(Fst = 0.003 – 0.076 within An. messeae,“ According to data on Table 3, this range of Fst should be 0.003 – 0.046.
Response
We changed this.
Comment 26
L 302. Change “inversions“ to “arrangements
Response
We changed this.
Comment 27
L 308. Use lower case in “20X”
Response
We changed this.
Comment 28
L 313. (L314) Change “The large increase in genomic differentiation“ to “A large increase in genomic differentiation”
Response
We changed this.
Comment 29
L 328. Change “The x-axis represents Fst“ to “The y-axis represents Fst”
Response
We changed this.
Comment 30
L 329. Change “the y-axis represents the genomic coordinates“ to “the x-axis represents the genomic coordinates”
Response
We changed this.
Comment 31
First paragraph of discussion should be rewritten to correct the following sentences: “Inversion X1 was found only as a fixed arrangement in messeae, whereas it was polymorphic in An. daciae.“ On the one hand, it is the chromosome that is or not is polymorphic, not a particular inversion. On the other hand, in the three localities An. messeae also shows the arrangement X4 in a non-negligible proportion ranging 2.6 - 6.4% (according to data in Table S1).
Response
We think that expression polymorphic inversion is common and largely used in literature.
Comment 32
“Another inversion, X4, was only found as a heterozygote in messeae.“ The expression “a heterozygote” seems to mean that it was only one mosquito with this karyotype in the samples, but Table S1 reveals that there were 11 mosquitoes carrying this arrangement. In addition, not all of them were heterozygotes: there are 9 females (heterozygotes) X14 and 2 males (hemizygotes) X4.
Response
We changed the sentence about X4 inversion as “Another inversion, X4, was only found as a heterozygote or hemizygote states in An. messeae.”
Comment 33
L 449. L451 Use italics “An. messeae”
Response
We changed this.
Comment 34
L 449. Change “Two species demonstrate differences “ to “These two species show differences”
Response
We changed this.
Comment 35
References should be carefully checked. The format of many citations must be corrected to standardize them. Many of the references from 28 to 77 have the title within square brackets [ ]. Response
We corrected the references.
Comment 36
Supplementary table S1.
“Letters M, W, and Y in genotypes indicate the presence of double picks in ITS2 sequences as follows: M – A and C, W – A and T, Y – T and C.” Could be changed by something like this “The presence of double picks in ITS2 sequences is indicated by the IUPAC nucleotide ambiguity code”. I do this comment because there are two more letters in the table (for the hybrid individual ME21; R – A and G, S – C and G). “Males can be recognized by a single inversion number in the X chromosome karyotype that corresponds to one homolog of this chromosome in males.” Here “inversion” should be changed to “arrangement” since 0 corresponds to the chromosome without inversions.
Response
We changed the title of supplementary table S1 as, “Supplementary table S1. ITS2 genotypes and inversion karyotypes in three populations of An. messeae s.l. in Moscow region. Letters in sample numbers indicate region (M – Moscow) and location (E – Yegoryevsk, N – Novokosino, No – Noginsk). The presence of double picks in ITS2 sequences is indicated by the IUPAC nucleotide ambiguity code: M – A and C, W – A and T, Y – T and C, R – A and G, S – C and G. Small letters indicate manually assigning base. Numbers in karyotypes represent standard (0) or inverted (1, 4) arrangements. The unique inversion on 2R chromosome arm is indicated as 4*. Males can be recognized by a single arrangement number in X chromosome karyotype that corresponds to one homolog of this chromosome in males.”
Reviewer 2 Report
See attachment

Author Response
Reviewer comments and our responses
Comment 1
In this paper the authors describe the relationship between two closely related anopheline mosquito species using a cytogenetic and genomic approach. These species are especially interesting because they appear to hybridize to some extent making them a potentially interesting system for further study on speciation.
Response
Thank you for the high evaluation of the significance of our study.
Comment 2
Figure 6 - Indicate the position of the inversion that distinguish An. messeae from An. daciae.
Response
We agree with the reviewer that it would be very interesting to see the differences in Fst values inside and outside of the inversions. However, this is currently impossible because An. messeae or An. daciae genomes have not been sequenced yet and there is no information about positions of the inversions in the genome.
Comment 3
Figure 7A - Distinguish the putative hybrids from parental An. messeae and An. daciae (e.g. use different colored dots).
Response
We changed the figure and indicated the hybrids on PCA plot.
Comment 4
Lines 413-415, The authors state “…the level of genomic divergence was quite low along the autosomal chromosomal arms…”. However, the plots depicted in Figure 6 show numerous peaks across both autosomes, these may represent regions under selection. The authors go on to suggest “…gene flow is still ongoing in most of the genome.” But the low Fsts may be the result of shared ancestral polymorphism, they should consider this in their discussion.
Response
We changed this sentence as “At the same time, the level of genomic divergence along the autosomal chromosomal arms was much lower than in chromosome X indicating that the gene flow is still ongoing in most of the genome or a high level of ancestral shared polymorphism is still present.”
Comment 5
Lines 433-434 “Such a high level of the genomic divergence on chromosome X between An. messeae and An. daciae could be explained by the inversions located on this chromosome” Include in your discussion mention of the “large X-effect” with respect to the role of the X chromosome in speciation. It would fit in well with your data. See Presgraves DC. Mol Ecol 2018 for a recent review.
Response
We added this citation to the discussion.
